# Disabled Homolog 2 (DAB2) Protein in Tumor Microenvironment Correlates with Aggressive Phenotype in Human Urothelial Carcinoma of the Bladder

**DOI:** 10.3390/diagnostics10010054

**Published:** 2020-01-20

**Authors:** Yoshitaka Itami, Makito Miyake, Sayuri Ohnishi, Yoshihiro Tatsumi, Daisuke Gotoh, Shunta Hori, Yousuke Morizawa, Kota Iida, Kenta Ohnishi, Yasushi Nakai, Takeshi Inoue, Satoshi Anai, Nobumichi Tanaka, Tomomi Fujii, Keiji Shimada, Hideki Furuya, Vedbar S. Khadka, Youping Deng, Kiyohide Fujimoto

**Affiliations:** 1Department of Urology, Nara Medical University, 840 Shijo-cho, Kashihara, Nara 634-8522, Japan; y.itami.324@gmail.com (Y.I.); makitomiyake@yahoo.co.jp (M.M.); sayuri3@naramed-u.ac.jp (S.O.); takuro.birds.nest@gmail.com (Y.T.); dgotou@gmail.com (D.G.); horimaus@gmail.com (S.H.); tigers.yosuke@gmail.com (Y.M.); kota1006ida@yahoo.co.jp (K.I.); kenzmedico0912@yahoo.co.jp (K.O.); nakaiyasusiuro@live.jp (Y.N.); you1513tt@yahoo.co.jp (T.I.); sanai@naramed-u.ac.jp (S.A.); sendo@naramed-u.ac.jp (N.T.); 2Department of Pathology, Nara Medical University, 840 Shijo-cho, Kashihara, Nara 634-8522, Japan; fujiit@naramed-u.ac.jp; 3Department of Pathology, Nara City Hospital, 1-50-1 Higashi kidera-cho, Nara 630-8305, Japan; k-shimada@nara-jadecom.jp; 4Division of Urology, Cedars-Sinai Medical Center, Los Angeles, CA 90048, USA; hfuruya@hawaii.edu; 5Bioinformatics Core, Department of Complementary and Integrative Medicine, University of Hawaii John A. Burns School of Medicine, Honolulu, HI 96813, USA; vedbar@hawaii.edu (V.S.K.); dengy@hawaii.edu (Y.D.)

**Keywords:** DAB2, bladder cancer, epithelial-mesenchymal transition

## Abstract

Disabled homolog-2 (*DAB2*) has been reported to be a tumor suppressor gene. However, a number of contrary studies suggested that DAB2 promotes tumor invasion in urothelial carcinoma of the bladder (UCB). Here, we investigated the clinical role and biological function of DAB2 in human UCB. Immunohistochemical staining analysis for DAB2 was carried out on UCB tissue specimens. DAB2 expression levels were compared with clinicopathological factors. DAB2 was knocked-down by small interfering RNA (siRNA) transfection, and then its effects on cell proliferation, invasion, and migration, and changes to epithelial-mesenchymal transition (EMT)-related proteins were evaluated. In our in vivo assays, tumor-bearing athymic nude mice subcutaneously inoculated with human UCB cells (MGH-U-3 or UM-UC-3) were treated by *DAB2*-targeting siRNA. Higher expression of DAB2 was associated with higher clinical T category, high tumor grade, and poor oncological outcome. The knock-down of DAB2 decreased both invasion and migration ability and expression of EMT-related proteins. Significant inhibitory effects on tumor growth and invasion were observed in xenograft tumors of UM-UC-3 treated by DAB2-targeting siRNA. Our findings suggested that DAB2 expression was associated with poor prognosis through increased oncogenic properties including tumor proliferation, migration, invasion, and enhancement of EMT in human UCB.

## 1. Introduction

Urothelial cancer of the bladder (UCB) is the second most frequent neoplasm of the urogenital organs [1] and is recognized to be a heterogeneous disease. Cancer-specific survival of patients with category Ta UCB is favorable, but T1 non-muscle invasive bladder cancer (NMIBC) has the potential to progress to muscle-invasive bladder cancer (MIBC) [2]. Cancer-specific survival in patients with MIBC is unacceptable despite treatments, including radical cystectomy with or without perioperative cisplatin-based chemotherapy [3]. To improve treatment results, the processes related to tumor growth, invasion, metastasis, and treatment resistance need to be explained. Tumor tissue consists of cancer cells and several types of stromal cells including macrophages, fibroblasts, and endothelial cells. Their interaction and crosstalk may influence the development of a cancer-related microenvironment for tumor growth and invasion.

With regard to tumor invasion, tumor budding is a pathological situation at the malignant invasion front in which each cancerous cell and/or small clusters of tumor cells invade the stromal area [4,5]. In human UCB, Jimenez et al. first evaluated the growth patterns at the tumor invasion front pathologically [6]. They categorized human invasive UCB into three groups: nodular, trabecular, and infiltrative. Tumor budding and the classifications of infiltrative growth are alike. The following studies revealed that this pathological parameter was significantly related to prognostic factors for both T1 tumors and MIBC [7,8,9,10,11]. In our previous study [12], pathological analysis using the orthotopic tumor model showed that MGH-U-3, UM-UC-14, and UM-UC-3, which are human urothelial cancer cell lines, display distinctive nodular, trabecular, and infiltrative patterns, respectively. As a result of comprehensive gene expression analysis by microarray (25 K Human Oligo chip, Toray Industries, Inc., Tokyo, Japan), we identified 16 genes with >2-fold upregulation in UM-UC-14 versus MGH-U-3, and more than 8-fold upregulation in UM-UC-3 versus UM-UC-14. One of the 16 genes which might potentially contribute to the formation of infiltrative patterns was disabled homolog-2 (*DAB2)*. 

The *DAB2* gene, initially named as *DOC-2* (deleted in ovarian 3 carcinoma 2) [13], is identified as one of two mammalian orthologs of the *Drosophila* disabled gene located on chromosome 5p13. Alterations in chromosome 5p13 have been recognized in bladder cancer specimens [14]. In earlier reports, *DAB2* was considered to be a tumor suppressor gene due to its low expression and loss in several tumor cells [15,16,17]. However, Chao et al. suggested that DAB2 downregulation by miR-187 can prevent cell migration, while the upregulation of DAB2 can stimulate epithelial-mesenchymal transition (EMT), leading to escalation of the capacities of tumor cell migration and invasion [18]. Two other groups have also described that DAB2 promotes tumor cell migration [19,20]. Heijden et al. reported that *DAB2* is a tumor progression gene in T1 grade 3 UCB [21]. 

DAB2 may exert numerous roles in tumor alteration and progression. The present study exposed that DAB2 expression was upregulated, along with the aggressive phenotype of human UCB, suggesting that DAB2 may play essential clinical and biological functions in UCB.

## 2. Materials and Methods 

### 2.1. IHC of Human Samples and Murine Xenograft Tumors

The Ethics Committee of the Nara Medical University accepted this study (the project identification code: 1256, the date of approval: 13th June 2016 ), and informed consent was provided by all patients. The study was conducted in compliance with the provisions of the Declaration of Helsinki (2013).

To investigate the expression levels of DAB2, IHC was performed. Paraffin-embedded tissues gained from all 212 patients at the initial transurethral resection of the bladder tumor. We used them to explore the relationship between DAB2 expression levels and clinicopathological variables. 

Immunohistochemical (IHC) staining using paraffin-embedded, formalin-fixed tissue blocks was performed as previously explained [11]. Slides were incubated overnight at 4 °C with anti-DAB2 antibody (sc13982, Lot#G0115; rabbit polyclonal, dilution 1/500; Santa Cruz Biotechnology, Santa Cruz, CA, USA). The slides were counterstained with hematoxylin and eosin (HandE), dried, and covered by a cover slide. IHC score was defined by assigning a population score and an intensity score. The population score was allocated, which represented the estimated population of positive-staining cells (0, none; 1, 0–25%; 2, 25–50%; 3, 50–75%; 4, 75–100%). The intensity score was allocated, which represented the average intensity of positive cells (0, none; 1, low; 2, intermediate; and 3, high). The population and intensity scores were combined to get a total score. We defined a total score ≤ 4 as low and ≥ 5 as high expression. We evaluated intravesical recurrence-free survival and progression-free survival in NMIBC patients and cancer-specific and overall survival in MIBC patients.

Resected tumors in a xenograft mouse model were examined by HandE and IHC staining analysis, as mentioned above. Anti-DAB2 antibody (1/500), anti-KRT14 antibody (HPA023040; rabbit polyclonal, dilution 1/2500; ATLAS ANTIBODIES, Voltavägen, Sweden), and anti-OCLN antibody (Occludin, HPA005933; rabbit polyclonal, dilution 1/500; ATLAS ANTIBODIES) were used for IHC analysis. 

### 2.2. Quantitative Reverse Transcription-Polymerase Chain Reaction (RT-qPCR) 

RNA extraction and RT-qPCR were conducted as previously reported [22]. Total RNA was extracted using the RNeasy Micro Kit (Qiagen GmbH, Hilden, Germany), and a QuantiTect Reverse Transcription Kit (Qiagen) was used in the synthesis of cDNA. Glyceraldehyde 3-phosphate dehydrogenase (GAPDH) was used as an internal control.

### 2.3. Microdissection of Cancerous and Stromal Tissues

For each HandE-stained sample, corresponding sections containing cancer lesions of interest were cut at 8-μm intervals to extract total RNA using the RNeasy FFPE kit (Qiagen). We separated unstained preparations from cancerous and stromal tissues by microdissection while comparing to HandE staining. Then, total RNA was extracted as mentioned above.

### 2.4. Dual Immunofluorescent Staining of Human Bladder Cancer Tissue

Dual immunofluorescent staining was performed using antibodies specific to DAB2 (rabbit monoclonal) or αSMA (A2547; mouse monoclonal, Sigma-Aldrich, St Louis, MO, USA). Frozen segments of human UCB tissue were fixed in pre-cooled (−20 °C) methanol for 5 min and blocked in 1% BSA for 1 h. The segments were incubated with anti-DAB2 (dilution, 1/200) and αSMA (dilution, 1/1000) for 1 h at 4 °C and rinsed three times in PBS. The segments were incubated in Alexa Fluor 594 anti-rabbit IgG (711-585-152; Jackson ImmunoResearch, Philadelphia, PA, USA) and Alexa Fluor 488 anti-mouse IgG secondary antibody (715-545-150). 

### 2.5. Cell Lines and Reagents 

In this study, human urothelial carcinoma cell lines MGH-U-3, UM-UC-3, J82, and T24 were used. MGH-U-3 was received from Dr. H. LaRue (Laval University Cancer Research Centre, Quebec, Canada). UM-UC-3, J82, and T24 were bought from the American Type Culture Collection (ATCC, Manassas, VA, USA). MGH-U-3 was derived from bladder cancer with pathological Ta (pTa) grade1, UM-UC-3 was pT2-4, the grade was unknown, J82 was pT3 grade3, and T24 was pTa grade3. In a normal moisturized incubator at 37 °C with 5% CO_2_, these cell lines were cultured in RPMI-1640 medium (Nacalai Tesque, Kyoto, Japan) added with 10% fetal bovine serum (FBS; JRH, Tokyo, Japan) and 1% penicillin and streptomycin (Thermo Fisher Scientific, Yokohama, Japan). Human bladder stromal fibroblasts (HBdSF) and specific media were purchased from ScienCell Research Laboratories, Carlsbad, CA, USA. HBdSF were plated in poly-L-lysine-coated culture vessels to promote cell attachment (ScienCell Research Laboratories, Carlsbad, CA, USA) and cultured in fibroblast medium (ScienCell Research Laboratories) in a normal moisturized incubator at 37 °C with 5% CO_2_.

### 2.6. Western Blot Analysis 

Western blotting (WB) was performed as previously described [11]. The primary antibodies employed were anti-DAB2 rabbit monoclonal antibody (dilution 1/500), anti-E-cadherin mouse monoclonal antibody (cat no. 3195; dilution 1/500, Cell Signaling Technology, Danvers, MA, USA), anti-N-cadherin mouse monoclonal antibody (cat no. 33-3900; dilution 1/500, Thermo Fisher Scientific), anti-vimentin rabbit monoclonal antibody (cat no. 5741; dilution 1/500, Cell Signaling Technology), anti-phospho-AKT rabbit monoclonal antibody (cat no. 4060; dilution 1/1000, Cell Signaling Technology), anti-phospho-ERK1/2 rabbit monoclonal antibody (cat no. 9101; dilution 1/1000, Cell Signaling Technology). Anti-actin mouse monoclonal antibody (clone AC-15; dilution 1/10,000, Sigma-Aldrich, Tokyo, Japan) was used as an inner loading control. Secondary antibody, horseradish peroxidase-conjugated goat anti-rabbit IgG (dilution 1/5000) or anti-mouse IgG antibody (dilution 1/10,000) (Santa Cruz Biotechnology), was incubated at 37 °C for 1 h.

### 2.7. Transfection of Small Interfering RNA (siRNA) 

*DAB2* siRNA (siRNA ID# s3896) and negative control siRNA were synthesized by Life Technologies (Tokyo, Japan). In a 6-well plate, UM-UC-3, J82, and T24 cells were transfected for 48 h with 25 pmol of siRNA and 3.75 μL of Lipofectamine™ RNAiMAX Transfection Reagent (Thermo Fisher Scientific) following the manufacturer’s instructions. RNA was extracted and expression of DAB2 was evaluated in each cell line. 

### 2.8. Cell Viability Assays 

To evaluate the effect of the degree of DAB2 expression, cell proliferation assays were performed. Three types of cells with different DAB2 expression (no treatment, negative control, or siRNA) were seeded into 96-well plates (2,000 cells/well) in serum-free medium, viable cell number after 12, 24, 48, and 72 h was measured by Cell Counting Kit-8 (Dojindo, Kumamoto, Japan), following the manufacturer’s instructions.

### 2.9. Matrigel Invasion and Migration Assay

We evaluated whether cell invasion and migration ability would change depending on the degree of DAB2 expression. For the Matrigel invasion assays, we used the BD Falcon Insert System (BD Biosciences, Piscataway, NJ, USA). The inject membrane was coated with 100 μL growth factor-reduced Matrigel (Corning Incorporated, Corning, NY, USA) for invasion assay or without Matrigel for migration assay. Three types of cells with different DAB2 expression (no treatment, negative control, or siRNA) were seeded in the upper chambers (2.0 × 10^4^ cells/well) in serum-free medium and incubated at 37 °C in an atmosphere of 5% CO_2_. At 48 h after incubation, the invading cells in the lower chambers were stained with Calcein AM (Promokine, Heidelberg, Germany), and the cells were directly calculated by a fluorescence microscope (Evos FL Auto, Life Technologies, Carlsbad, CA, USA). 

### 2.10. Interactive Co-Culture of Urothelial Cancer Cells and Bladder Stromal Fibroblasts 

MGH-U-3 has no endogenous DAB2 expression, whereas HBdSF does. These cells were interactively co-cultured on NICO-1 plates (Ginrei Lab Inc, Ishikawa, Japan). NICO-1 plates are horizontal co-culture plates in which the humoral factor can move freely through the filter but the cells cannot [23]. We prepared two types of HBdSF cells transfected with negative control siRNA or *DAB2* siRNA. MGH-U-3 and HBdSF (negative control siRNA or *DAB2* siRNA) seeded at the same cell density (5 × 10^4^ cells/well) were incubated in each plate for 3 days. After incubation, RT-qPCR of MGH-U3 was performed to investigate changes in EMT gene expression, as described below. 

### 2.11. RT² Profiler™ PCR Array

To identify the effects of DAB2 on gene expression in EMT, the Human Epithelial to Mesenchymal Transition RT² Profiler™ PCR Array (PAHS-090Z, Qiagen) was used. This PCR array has a total of 84 EMT-related gene primer sets and housekeeping genes in a 96-well plate. After DNase treatment and RNA clean up, the isolated RNA was reverse transcribed into cDNA following the manufacturer’s procedure. Real-time PCR Array was performed by the Super Array PCR master mix on an ABI StepOne Plus (The Lab World Group, Woburn, MA, USA). Data were analyzed using the ΔΔ*C*t method.

### 2.12. Animals

Animal care complies with the proposals of The Guide for Care and Use of Laboratory Animals (National Research Council of Japan). The animal facility committee at Nara Medical University (Nara, Japan) permitted this research (protocol ID: 12360, the date of approval: 2nd August 2018). Male 6–8 weeks old athymic BALB/c nu/nu mice were bought from Oriental Bio Service (Kyoto, Japan). All mice were raised under pathogen-free environments and supplied with sterile water and food.

### 2.13. Xenograft Model and Intratumoral Treatment 

After adapting the mice for a week in our facility, each mouse was injected into the flank with bladder cancer cells (UM-UC-3; 2.0 × 10^5^ cells/tumor, MGH-U-3; 5.0 × 10^5^ cells/tumor) in 50 μL RPMI-1640 medium plus 50 μL of Matrigel (Corning Inc.,Corning, NY, USA). When the tumor diameter reached 5 mm at 2 weeks after cell injection, mice were randomly divided into three groups: control (no treatment), negative control siRNA (10 μg of non-target siRNA mixed with 1.2 μL of in vivo-jetPEI reagent with an N/P ratio of 6), and human *DAB2* siRNA (10 μg of siRNA with 1.2 μL of in vivo-jetPEI reagent with an N/P ratio of 6). The number of mice in each group of control, negative control siRNA, and human DAB2 siRNA were three, five and six in the UM-UC-3 group, and three, five and five in the MGH-U-3 group. Then, we administered DAB2 siRNA intratumorally twice a week for 2 weeks. In vivo-jetPEI reagent (Polyplus-transfection Inc., New York City, NY, USA) was administered in combination with the siRNA to certify optimal delivery to xenograft tumors [24]. Diameters of the tumor were calculated once a week using electronic calipers, and the volumes of tumors were evaluated according to the formula: {(width)^2^ × length}/2 (mm^3^). After the final dose, all mice were euthanized by exsanguination after isoflurane inhalation and tumors were resected for subsequent examination.

### 2.14. Statistical Analysis 

We used GraphPad Prism 5.0 (GraphPad Software, Inc., San Diego, CA, USA) for statistical analysis and drawing diagrams. For statistical analysis, Student’s *t*-test or Mann–Whitney *U* test was applied as necessary. Survival curves were achieved by the Kaplan-Meier method and each prognostic factor was compared by the log-rank test. *p*-values < 0.05 were considered to be statistically significant.

## 3. Results

### 3.1. Association between DAB2 Expression and Clinicopathological Variables in Human Bladder Cancer 

Figure 1A shows representative IHC images of low, intermediate, and high expression of DAB2 in human UCB specimens. The predominant locations of DAB2 expression were in the cytoplasm of cancer and stromal cells. We semi-quantified DAB2 expression levels in cancer and stromal cells adjacent to the cancerous area separately. Table 1 presents the clinicopathological backgrounds of 154 patients with NMIBC, and of 58 patients with MIBC. Clinicopathological variables were compared with DAB2 expression in cancerous areas (high and low expression levels). The pathological characteristics, including clinical T stage, tumor grade, the presence of carcinoma in situ, histological variants and tumor infiltrative patterns, were significantly different between patients with low and high DAB2 expression. Similar results were observed in stromal areas (Appendix A).

In human UCB tissue specimens, we performed microdissection and quantified mRNA levels in cancer and stromal tissues separately. The results showed that DAB2 expression as quantified by IHC analysis was consistent with that by RT-qPCR analysis (mRNA). High expression of DAB2 in stromal tissues, as detected by immunostaining, was also observed at high mRNA levels in RT-qPCR (Figure 1B).

### 3.2. Prognostic Value of DAB2 Expression in Human Bladder Cancer

In the follow-up period, 35 (16.5%) of 212 patients died and 24 (11.3%) died of UCB at a median 55 months after initial TURBT. Of these 24 patients, 11 were initially diagnosed as NMIBC and then progressed to MIBC. The residual 13 patients were initially diagnosed as MIBC, had a radical cystectomy, and then relapsed. Of the 154 NMIBC patients, 18 (11.7%) had progressed to MIBC at a median 44 months. In contrast, 16 (27.6%) of the 58 MIBC patients had relapsed at a median 7 months after radical cystectomy, 13 (22.4%) died of UCB at a median 20 months after cystectomy. 

In NMIBC, high expression of DAB2 in cancer cells was not associated with intravesical recurrence-free survival, but 11 was associated with shorter progression-free survival (*p* = 0.97 and *p* = 0.027, Figure 2A). On the other hand, high expression of DAB2 in stromal cells was associated with shorter progression-free interval and intravesical recurrence-free survival (*p* = 0.011 and *p* = 0.008, respectively, Appendix A). In MIBC, high expression of DAB2, in both cancer and stromal cells, was associated with a shorter over-all and cancer-specific survival (*p* = 0.026, Figure 2B; *p* = 0.028, Appendix A; *p* = 0.008, Figure 2C; *p* = 0.007, Appendix A, respectively). When we stratified into three groups depending on the number of cancer and stromal cells highly expressing DAB2, overall and cancer-specific survival were significantly worse as the number of cells highly expressing DAB2 increased (*p* = 0.013, Appendix A and *p* = 0.012, Appendix A).

### 3.3. Evaluation of DAB2 Expression in Stromal Areas Adjacent to Cancerous Areas

To quantify DAB2 levels in cancerous and stromal areas adjacent to cancerous regions, dual immunofluorescent staining was performed with antibodies specific to DAB2 or αSMA. αSMA is specifically expressed in myofibroblasts in stromal areas and may be able to differentiate between cancerous and stromal areas. Representative images from fluorescent microscopy showed that DAB2 was expressed not only in UCB tissue but also in stromal tissue (Appendix A).

### 3.4. DAB2 Stimulates Tumor Migration and Invasion of Urothelial Cancer Cells 

WB and RT-qPCR analyses were done to prove the expression level of DAB2 in the four urothelial cancer cell lines. Three cell lines, except MGH-U-3, had endogenous DAB2 expression at the protein and mRNA levels (Figure 3A). WB analysis confirmed that DAB2 protein expression was knocked-down by siRNA transfection (Figure 3B). In addition, to investigate whether DAB2 downregulation was involved in EMT and the ERK1/2 and AKT pathways, WB was performed. The levels of vimentin expression and pMAPK/ tMAPK in UM-UC-3 and J82 cells transfected with *DAB2* siRNA were decreased compared with negative control group cells. Both UM-UC-3 and T24 cells showed a decrease of pAKT/ tAKT in DAB2 siRNA (Figure 3B). Downregulation of DAB2 significantly decreased migration and invasion ability and tended to decrease proliferative ability (Figure 4A–C). These results indicated that migration and invasion ability was enhanced by treatment with DAB2 in vitro.

We analyzed the genes whose expression level changed at least 2-fold in UM-UC-3 vs. MGH-U-3 and the corresponding log2 fold change values using Ingenuity Pathway Analysis (IPA) software (http://www.ingenuity.com). As a result of the core analysis function of IPA, we have revealed 22 genetic networks from our database including 279 differentially expressed genes in UM-UC-3 versus MGH-U3. We identified only networks involving DAB2 related to biological damage and abnormalities, connective tissue disorders, and cancer from these 22 networks (Appendix A). The network included a signal transduction pathway (AKT) and other cytoskeletal genes (*MYO6*, *KRT4* and Cytokeratin). This network also included EMT related proteins (Vimentin) and micro RNA-124.

### 3.5. DAB2 Produced from Bladder Stromal Fibroblasts Correlates with Promotion of EMT in Human UCB

To investigate the effect on EMT in UCB cells by DAB2 produced from bladder stromal fibroblasts, we used MGH-U-3 as human UCB cells that have no endogenous DAB2 expression, and HBdSF for human stromal fibroblasts. HBdSF has endogenous DAB2 expression, as we confirmed by WB analysis (Figure 5A). We knocked-down DAB2 expression by siRNA transfection (Figure 5A). We used interactive co-culture dishes (Nico-1^®^) which cannot pass cells: only fluid factors can pass through the filter freely (Figure 5B). Interactive co-culture of MGH-U-3 and HBdSF transfected with negative control siRNA or HBdSF transfected with DAB2-targeting siRNA was performed. After 48 h, changes to EMT-related genes in MGH-U-3 were evaluated by RT2 Profiler PCR assays. 

Nine genes were altered more than 2-fold in MGH-U-3 with HBdSF transfected with DAB2-targeting siRNA as compared to MGH-U-3 cultured with HBdSF transfected with negative control siRNA (Figure 5C). These results suggested that certain interactive humoral proteins were released from HBdSF cells with DAB2 expression and that these factors may contribute to EMT promotion in MGH-U-3 cells.

### 3.6. DAB2 siRNA Treatment Inhibits Tumor Growth and Invasion In Vivo 

An animal experimental process chart is shown in Figure 6A. All treatments were well tolerated without serious toxicity and body weight loss in our in vivo studies. Mice with UM-UC-3 tumors treated with DAB2 siRNA had significantly suppressedtumor growth during treatment. Figure 6B presents representative images of xenograft tumors for each group during tissue harvest. Among all treatment groups, UM-UC-3 xenograft treated with siRNA exhibited the smallest tumors. Tumor weights were significantly lower in mice treated with DAB2 siRNA compared to “no treatment” and negative control siRNA groups. However, in the xenograft model of MGH-U-3, there were no significant changes in tumor weight and tumor growth compared with the ‘no treatment’ and non-target siRNA groups (Figure 6C). In HandE stain, tumor invasion is observed in the control group mice, but not in the treatment group. By IHC, DAB2 and KRT14 were strongly expressed in control group cells, but OCLN was more weakly expressed than treatment group cells (Figure 6D).

### 3.7. DAB2 siRNA Treatment Inhibits EMT of Bladder Cancer Cells 

From the genes of EMT-related markers we detected in PCR assays, we selected two genes, KRT14 which is known as an EMT-promoting gene, and OCLN, which is known as an EMT down-regulating gene. Figure 6D demonstrates typical images of xenograft tumors stained by each marker in the UM-UC-3 group. By IHC investigation, DAB2 siRNA-treated mice xenografts revealed significantly lower expression of DAB2 compared to no treatment mice. Furthermore, infiltration around tumors treated with DAB2 siRNA was less than that with no treatment. DAB2 siRNA-treated mice xenografts revealed significantly lower expression of KRT14 and higher expression of OCLN in comparison with no treatment mice. Along with the knock-down of DAB2, decreased KRT14 was observed, whereas increased OCLN was seen. There was compatibility between these results and those of in vitro investigation. 

## 4. Discussion

DAB2 has been considered to be a tumor suppressor gene because DAB2 is lowly expressed in several tumor cells and is thought to inhibit tumor metastasis [15]. However, this study showed the opposite role for DAB2 during tumor progression. These diversities among cancer types and cell lines may be correlated with the tissue-specific differentially expression patterns of DAB2 [15]. DAB2 has two different splicing formats and encodes two isoforms (p96-DAB2 and p67-DAB2) [13]. Xu et al. reported that there are different functions between p96-DAB2 and p67 DAB2 in the process of oncogenesis on lung cancers [25]. UC cell lines in this study expressed mainly p96-DAB2 but depending on cancer types and cell lines, DAB2 expression isoforms and the location of expression differ, which is presumed to result in the difference of tumor promotion or suppression. In this study, DAB2 expression levels of MGH-U-3 cells with low malignant potential were significantly lower than J82 cells with high malignant potential (Figure 3A). DAB2 expression levels were also related to the clinical T stage and infiltration pattern in clinical UCB samples (Table 1 and Appendix A). IHC analysis revealed that DAB2 expression increased significantly with UCB progression (Figure 1A). This result raised a significant view that DAB2 may affect UCB progression and may have multiple functions related to tumor progression and transformation.

Cancer cell migration and invasion are the most important events leading to cancer progression and metastasis. This in vitro research showed evidence that DAB2 knock-down can inhibit cancer migration in UM-UC-3 cells and T24, and cancer invasion in J82 and T24 (Figure 4B,C). In the in vivo study, we tried to establish a subcutaneous tumor model using J82, but the tumor growth was too rapid to be suitable as a xenograft model, so we used UM-UC-3. Treatment of DAB2 knock-down in subcutaneous tumors in UM-UC-3 cells proved that DAB2 affects UCB cell growth and progression in vivo (Figure 6B,C). Targeting DAB2 can partly inhibit UCB progression. Therefore, it is necessary to determine the transcriptional controlling mechanisms involved in DAB2 expression.

DAB2 is needed for regulatory numerous signaling pathways, including the Wnt signaling pathway and Ras/MAPK pathway as an essential adaptor molecule [26]. DAB2 has been reported as a target gene of transforming growth factor-β (TGF-β) and also affects EMT [27]. EMT allows tumor cells to gain a migratory and invasive ability, leading to tumor progression [28]. Chao et al. suggested that miR-187 can down-regulate DAB2 and suppress cell migration, while the upregulation of DAB2 can stimulate EMT, resulting in enhanced tumor cell migration and invasive capacity [18]. In addition, DAB2 is a mitogen-responsive phosphoprotein that can interrelate with Grb2 and regulate RAS pathways and growth factors [29]. 

This result revealed that DAB2 plays an important role in the progression of UCB, but the basic molecular systems related to migration and invasion by DAB2 remain unknown. It has been reported that DAB2 can regulate tumor progression by stimulating EMT-dependent metastasis [30]. In this study, DAB2 knock-down in J82 cells, which produce large amounts of endogenous DAB2, contributed to decreases in vimentin and pMAPK concentrations (Figure 3B). Furthermore, high expression of DAB2 in stromal lesions indicated poor prognosis in human UCB samples. Fibroblast cells are representative of stromal tissue, HBdSF is a human bladder fibroblast cell line in which we demonstrated endogenous DAB2 expression by WB (Figure 5A). In vitro, humoral factors of HBdSF affected EMT promotion in MGH-U-3 cells, which has no endogenous DAB2 (Figure 5C). Murine xenografts administered with D*AB2* siRNA displayed significantly lower expression of *KRT14* and higher expression of *OCLN* compared with ‘no treatment’ (Figure 6D). 

It has been reported that cancer-associated fibroblasts (CAFs) affect the inflammatory environment around tumors [31]. CAFs secrete chemokines, growth factors, and extracellular matrix elements including collagen I/IV, fibronectin [32]. A recent report has shown that a hypoxia condition and subsequent elevated HIF-1α levels increase fibrogenesis, leading to stimulate the formation of collagen I and fibronectin by fibroblasts in pancreatic cancer cells [33]. This mechanism facilitates a close tumor-promoting relationship between cancer and stromal cells. Based on our results, we present the mechanism by which DAB2 has a serious impact on human UCB. DAB2 expression in human UCB is particularly observed in high-grade and high-stage tumors and stromal areas. The role of DAB2 is that expression in CAFs promotes EMT of tumor cells, and induces tumor migration and invasion, resulting in the accumulation of cancer cells that are beneficial to the tumor microenvironment. Thus, the DAB2-related pathway in the tumor microenvironment may be involved in disease progression. Inhibiting this pathway may be an alternative approach to existing UCB treatment, and may contribute to improving cancer prognosis.

This study had some limitations. Firstly, we only evaluated fibroblasts from stromal tissue in vitro. It is also necessary to consider the possibility that other stromal cells, such as immune cells, pericytes, endothelial cells, and inflammatory cells, could also affect tumor invasion. Secondly, this study did not evaluate cytokines that affect cell apoptosis, proliferation, and EMT. To elucidate the relationship between DAB2 and tumor cells, cytokines, and the microenvironment surrounding the cancer cells, we need further experimentation. Recent reports have shown that nerve growth factor (NGF) promotes EMT in prostate cancer cells via activation of the NGF tyrosine kinase receptor, tropomyosin receptor kinase A [34]. Lastly, we had variations related to tumor injection, tumor measurement, and intratumoral administration that might affect the outcomes of our in vivo study.

## 5. Conclusions

DAB2 expression both in cancerous and stromal lesions was associated with poor prognosis through increased oncogenic properties including tumor proliferation, migration, and invasion in human UCB. Furthermore, our in vitro study showed that DAB2 expression of fibroblasts in stromal lesions may affect the promotion of EMT in cancer cells. Inhibiting the role of DAB2 can lead to tumor shrinkage and suppression of invasion, which may help improve the prognosis of UCB in clinical practice.

## Figures and Tables

**Figure 1 diagnostics-10-00054-f001:**
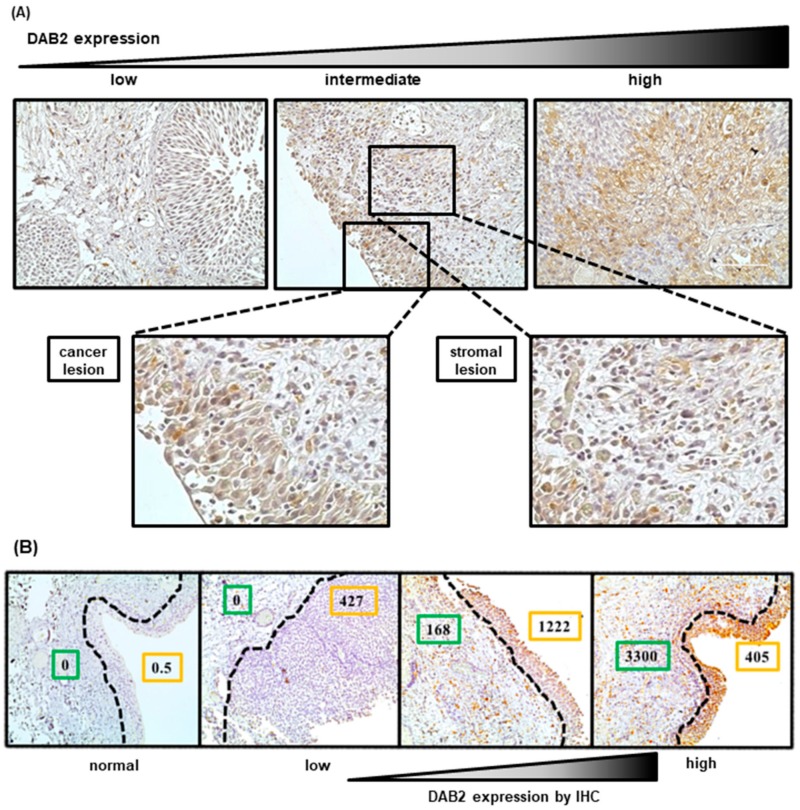
(**A**) Immunohistochemical staining for Disabled homolog-2 (DAB2) revealed that DAB2 was expressed mainly in the cytoplasm of cancer and stromal cells. Scale bars, 200 μm. (**B**) *DAB2* mRNA expression levels as determined by RT-qPCR in cancer and stromal tissues separated by microdissection. The stronger the immunohistochemical staining of DAB2, the higher the *DAB2* mRNA expression in cancer and stromal tissues, respectively. Green line: stromal lesion, orange line: cancerous lesion, Dotted line: resection line of microdissection, IHC: Immunohistochemical staining.

**Figure 2 diagnostics-10-00054-f002:**
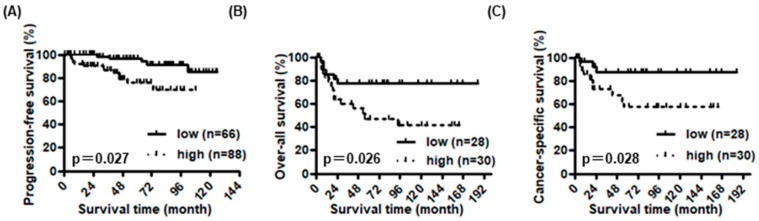
The association between clinicopathological information of the 154 non-muscle invasive bladder cancer (NMIBC) patients and 58 muscle-invasive bladder cancer (MIBC) patients and their DAB2 expression in cancer lesions. In NMIBC, there was a significant association between DAB2 expression and progression-free survival (**A**). In MIBC, over-all survival (**B**) and cancer-specific survival (**C**) were significantly lower in patients with high DAB2 expression than that in patients with low DAB2 expression.

**Figure 3 diagnostics-10-00054-f003:**
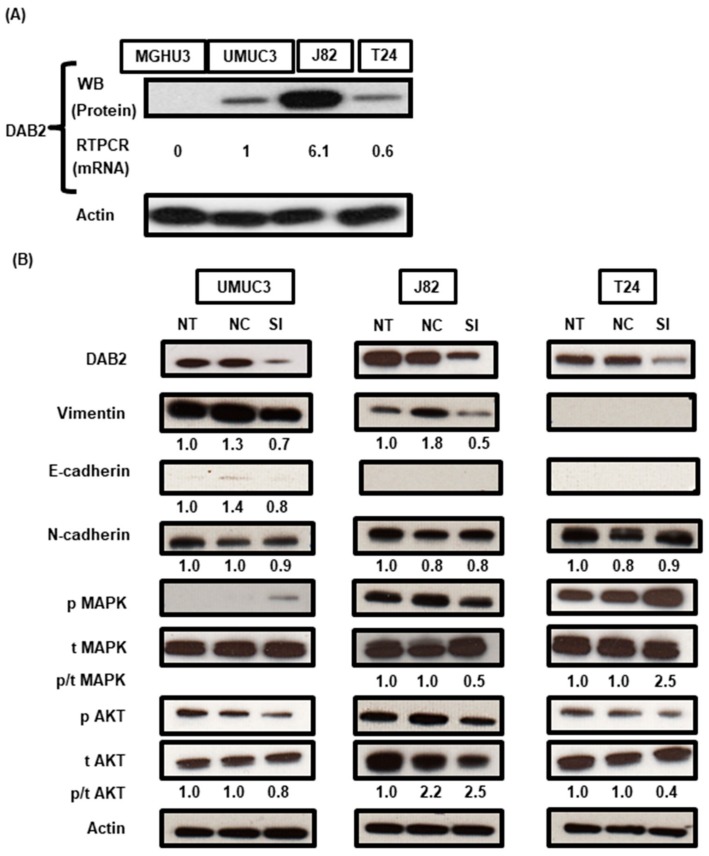
(**A**) In four human UC cell lines, western blot analysis (WB) revealed DAB2 protein expression in UM-UC-3, J82, and T24 cells, whereas MGH-U-3 cells did not express DAB2 protein. The RT-qPCR analysis also revealed mRNA expression of *DAB2* corresponded to the results of WB. (**B**) In three cell lines knocked-down with *DAB2* siRNA transfection, DAB2 protein reduction was confirmed by WB. Epithelial-to-mesenchymal transition-related markers, proteins involved in the activation of intracellular signaling pathways, were quantified by western blot analysis. Actin was used as a loading control. The numbers below the bands are the results of densitometric quantification. NT: No treatment; NC: Negative control; SI: *DAB2*-si RNA.

**Figure 4 diagnostics-10-00054-f004:**
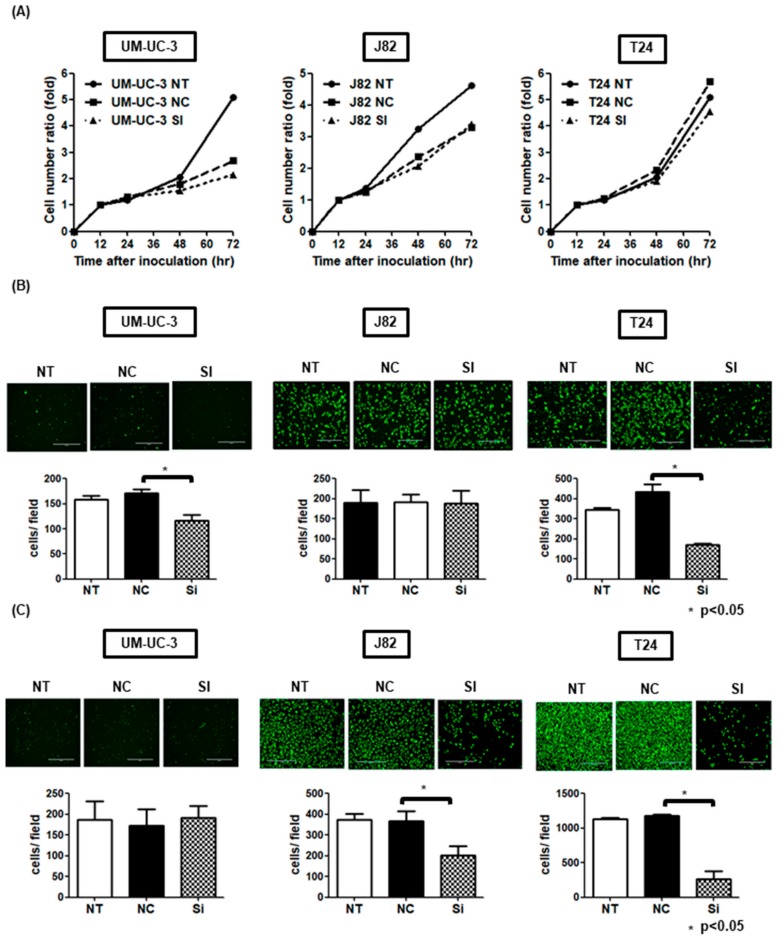
(**A**) In cell proliferation assays, the proliferation ability of UM-UC-3 NT tended to be higher than that of NC and SI compared to T24. But there were no significant differences in the proliferative ability for all three cell lines. (**B**) In migration assays, there was significant decreased migration ability of si-RNA transfected cells than negative control cells in UM-UC-3 and T24. (**C**) In the Matrigel invasion assay, there was significant decreased invasion ability of si-RNA transfected cells than negative controls in J82 and T24. NT: No treatment; NC: Negative control; SI: *DAB2*-si RNA.

**Figure 5 diagnostics-10-00054-f005:**
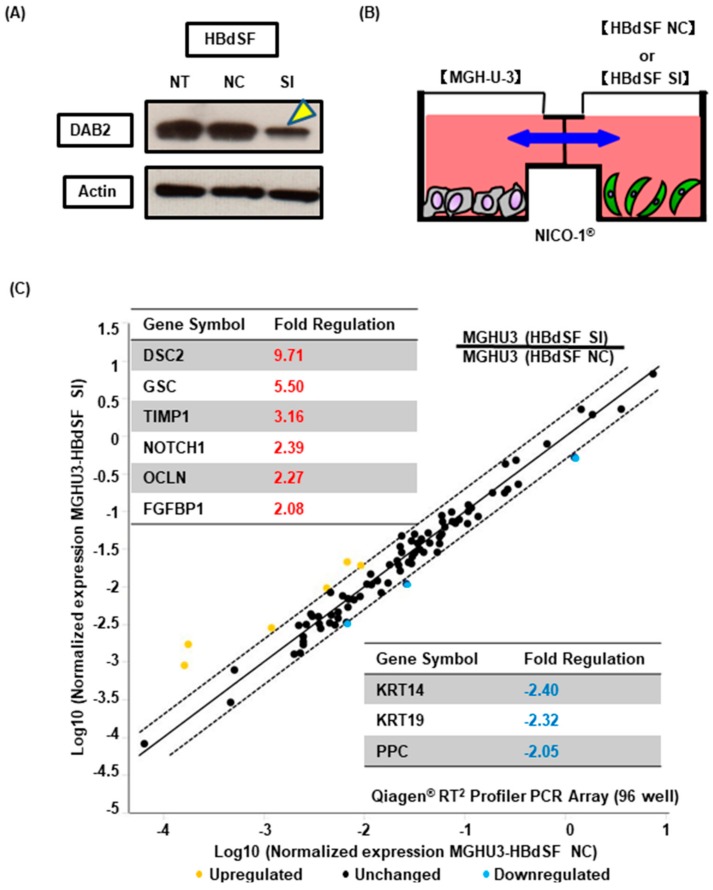
(**A**) HBdSF cells knocked-down by DAB2 siRNA transfection, DAB2 protein reduction was confirmed by WB. Yellow arrowhead indicates DAB2 protein bands reduced by siRNA. (**B**) Indirect co-culture dish (Nico-1^®^) cannot pass cells, whereas only fluid factors can pass through the filter freely. We placed MGH-U-3 cells in one plate and HBdSF negative control or si RNA cells in another plate. (**C**) Changes in epithelial-to-mesenchymal transition (EMT) markers in MGH-U-3 cells were evaluated by PCR assays for EMT markers. NT: No treatment; NC: Negative control; SI: *DAB2*-si RNA.

**Figure 6 diagnostics-10-00054-f006:**
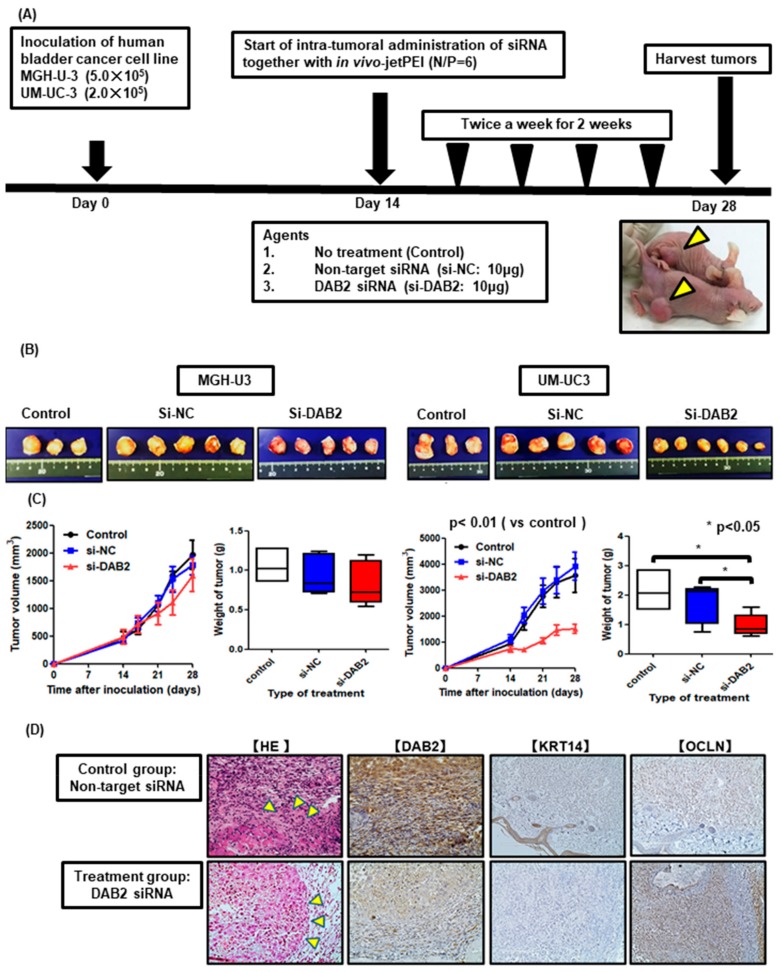
(**A**) Schematic drawings showing this experiment workflow. We injected mice with UM-UC-3 cells (2 × 10^5^/tumor) and MGH-U-3 cells (5 × 10^5^/tumor) together with Matrigel and divided them into three groups (*n* = 3; control (no treatment), *n* = 5; negative control siRNA, and *n* = 5 or 6; human *DAB2* siRNA) after two weeks. Then, we treated mice twice a week for 4 weeks. Three days after the last treatment, we euthanized mice and harvested xenografts. Yellow triangles show subcutaneous tumors. (**B**) These pictures were xenografts resected from treatment groups. (**C**) Tumor growth rate during treatment was significantly lower in UM-UC-3 mice treated with *DAB2* siRNA compared to both the control and negative control siRNA group (Mann–Whitney U test; * = *p* < 0.05). Intratumoral treatment with *DAB2* siRNA resulted in a significant weight loss of excised xenografts compared to both the control and negative control siRNA group (Mann–Whitney U test; * = *p* < 0.05). However, in MGH-U-3 mice, both tumor growth rates and resected tumor weights were not significantly different between groups. (**D**) Representative images of xenograft tumors stained for each marker in UM-UC-3 group cells. Yellow triangles indicate tumor infiltration in control group mice but not in treatment group animals.

**Table 1 diagnostics-10-00054-t001:** Clinicopathological characteristics of 212 patients with bladder cancer undergoing transurethral resection of the bladder tumor stratified by DAB2 expression level in the cancerous area.

		Number of Patients (%)	DAB2 Expression		*p*-Value †
			Low	High	
Total		212	116	96	
Age, years, median (range)		71 (34–94)	70 (34–94)	71 (41–93)	0.64
Gender	Male	181 (86.8)	86	95	0.19
	Female	28 (13.2)	17	11	
clinical T stage	Ta	68 (32.1)	48	20	0.014
	T1	81 (38.2)	39	42	
	Tis	13 (6.1)	5	8	
	≥T2	50 (23.6)	24	26	
Grade	Low	71 (33.5)	55	16	<0.001
	High	141 (66.5)	61	80	
Carcinoma in situ	Negative	137 (64.6)	91	46	<0.001
	Positive	75 (35.4)	25	50	
Lymphovascular invasion (≥T1)	Negative	65 (49.6)	37	28	0.045
	Positive	66 (50.4)	26	40	
Histological variants	Yes	33 (15.6)	11	22	<0.01
	No	179 (84.4)	105	74	
Lymph node status	N0	207 (97.6)	114	93	0.83
	N1, N2	5 (2.3)	2	3	
Tumor size	≤3 cm	150 (70.8)	85	65	0.38
	>3 cm	62 (29.2)	31	31	
Number of tumor	single	122 (57.5)	70	52	0.36
	≥2	90 (42.5)	46	44	
Infiltration pattern (INF)	a	86 (40.6)	57	30	0.028
	b	83 (39.2)	40	42	
	c	43 (20.2)	19	24	

† comparing age uging t test and other two groups using the Chi square test.

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
