# Peer review of "Disabled Homolog 2 (DAB2) Protein in Tumor Microenvironment Correlates with Aggressive Phenotype in Human Urothelial Carcinoma of the Bladder"

_diagnostics, 2020, doi:10.3390/diagnostics10010054_

Round 1

Reviewer 1 Report

The paper suggests that DAB2 is a tumor promoter and not a tumor suppressor as is commonly reported.  The results seem sound based strictly on the internal contents.  The problems start with the paper which presents the results prompting this study (Miyake et al 2017).  In that report, 15 other candidates are shown, 2 of which were examined.  There, both selected gene products had some impact on bladder cancer.  How did DAB2 stand out to be the next selected for study?  Several other genes on that list could have been selected.  How should I be convinced that the DAB2 results are independent of the demonstrated impact of Col13A1 and/or COL4A1 or the possible impact of several other cancer-related genes on the list?

The data presented should report on several of the other genes from the list.

This point is further raised by the contrary findings.  The authors should consider obtaining cancer samples from outside the major demographic represented by the current population to see if there is a different outcome.

There is a suggestion that the impact of DAB2 over-expression is on EMT.  The results shown on this  point are not compelling.

T24 and UM-UC-3 should be included in the results shown in Fig 3C so we can have a comparative look for results shown in figs 4B and C.  Figure 6D does not convey any idea how extensive the infiltration was among the tumors and the fact that the tumors could isolated as shown in 6B suggest that this was not a major issue.

Reviewer 2 Report

This is an interesting and timely manuscript on the role of DAB2 in tumorigenesis. Although generally considered as a tumor suppressor in bladder cancer, there are evidences for an oncogenic role of DAB2  in other cell systems. Here, the authors give experimental evidence for a DAB2 role as a tumor promoter in bladder cancer. This includes not only expression data but also functional studies that make the conclusions more sound. In general is a good work and I only have minor concerns to improve the presentation and readability of the results. Lastly, I would like to see some comparative discussion on the new role of DAB2 as tumor promoter vs. the first role as tumor suppressor. What are the differences with previous works?

Minor points.

-Figure 1B/panel "high": Is quantification correct in the cancerous lesion?

-Figure 2 should be improved. It is difficult to distinguish among patients with low/high expression of DAB2 (especially in panel 2A).

-Figure 3A: A beta-actin control is required!!!!! Controls are always required!!. For the qPCR panel units (1, 6.1, 0.6) should be given

-Figure 4A: This reviewer sees differences in proliferation among the three cell lines (compare UM-UC-3 with T24). The text should be corrected. Furthermore the figure would be more easily understandable if the definition for the Y-axis is made more clear

-Figure 6C: this figure would be more easily understandable if the panels were a little bit more separated one each other

Reviewer 3 Report

The manuscript presented by Dr Itami and colleagues is focused on the role of DAB2 in urothelial carcinoma of the blad.

The authors use different systems, in particular, they work with cells in vitro, with mice and have also an analysis of patients.

The manuscript is interesting, the pictures are well performed but other experiments are required for the final submission.

Furthermore, in the discussion, the authors add some conclusions that don't derive from their data.

Revision:

-Paragraphs 2.4 and 2.6: The authors should add the antibodies codes. lines -178-179. The authors should explain how they have performed migration assay.

-line 177: how were the cells calculated with fluorescence microscopy?

-187: what about the primers used for investigating the changes in EMT gene expression?

-In fig. 3A the cells employed express different levels of DAB2. Also if the authors add some information about the cells used in the discussion, it could be better to add information in previous sections.

-In fig. 3C the authors consider only vimentin and N-cadherin that are bot markers of a mesenchymal phenotype. While the changes in vimentin levels expression are clearer after siRNA approaches, it is not the same about N-cadherin expression. The authors should also investigate the reciprocal and opposite trend of E-cadherin and/or cytokeratin levels.

-In fig. 3 C probably a fold rate increase between pAKT/AKT levels could be appreciated.

-In fig. 3C cells invasion is not visible in UM_UC-3 cells under any conditions. Why did the authors choose just these cells for in vivo experiments with xenografted mice? They should spend a few words for discussing the in vitro VS in vivo results (also relating to lines 382-383)

-Line 405: HBdSF cells are positive for alpha SMA and Fap-1 expression for example? They represent reactive stroma or not? If the authors have not this evidence it could be better do not address confounding information.

- To the end in the discussion section, authors should amplify the discussion section about EMT, also considering the recent pieces of evidence obtained in other cell cancer types (e.g https://doi.org/10.3390/cancers11060784)

Round 2

Reviewer 1 Report

The paper is ready for publication.  I recommend one minor improvement.  In Figure 5 the yellow color of the fold-regulation numbers does not contrast well over the gray.  Red would be a better color for the font.

Reviewer 3 Report

For me it is OK in this form

This manuscript is a resubmission of an earlier submission. The following is a list of the peer review reports and author responses from that submission.